# Comparative genomics of Tn*6411* transposons carrying the *bla*$_{IMP-1}$ gene in *Pseudomonas aeruginosa*

Lin Zheng[1]*, Zixian Wang[1], Jingyi Guo[2], Jiayao Guan[3], Gejin Lu[1], Jie Jing[1], Shiwen Sun[1], Yang Sun[1], Xue Ji[1], Bowen Jiang[1], Yongjie Wang[4], Chuanfang Zhao[5], Lingwei Zhu[1]*, Xuejun Guo[1]*

1 Changchun Veterinary Research Institute, State Key Laboratory of Pathogen and Biosecurity, Key Laboratory of Jilin Province for Zoonosis Prevention and Control, Chinese Academy of Agricultural Sciences, Changchun, China, 2 The Second Clinical Medical College of Jilin University, Changchun, Jilin, China, 3 College of Veterinary Medicine, Jilin Agriculture University, Changchun, Jilin, China, 4 Department of Spinal Surgery, The First Hospital of Jilin University, Changchun, Jilin, China, 5 Institute of Special Animal and Plant Science of Chinese Academy of Agricultural Sciences, Changchun, Jilin, China

* xuejung2021@163.com (XG); lingweiz@126.com (LZ); zl050514@163.com (LZ)

**Data Availability Statement:** The contig sequences of strains 18081308 and 18083286 have been submitted to GenBank under accession numbers GCA_024718375.1 and

## Abstract

We aimed to determine the molecular characteristics of carbapenem-resistant *Pseudomonas aeruginosa* strains 18081308 and 18083286, which were isolated from the urine and the sputum of two Chinese patients, respectively. Additionally, we conducted a comparative analysis between Tn*6411* carrying *bla*$_{IMP-1}$ in strain 18083286 and transposons from the same family available in GenBank. Bacterial genome sequencing was carried out on strains 18081308 and 18083286 to obtain their whole genome sequence. Average nucleotide identity (ANI) was used for their precise species identification. Serotyping and multilocus sequence typing were performed. Furthermore, the acquired drug resistance genes of these strains were identified. The carbapenem-resistant *P. aeruginosa* strains isolated in the present study were of sequence type ST865 and serotype O6. They all carried the same resistance genes (*aacC2*, *tmrB*, and *bla*$_{IMP-1}$). Tn*6411*, a Tn*7*-like transposon carrying *bla*$_{IMP-1}$, was found in strain 18083286 by single molecule real time (SMRT) sequencing. We also identified the presence of this transposon sequence in other chromosomes of *P. aeruginosa* and plasmids carried by *Acinetobacter spp.* in GenBank, indicating the necessity for heightening attention to the potential transferability of this transposon.

## Introduction

*Pseudomonas aeruginosa* is a zoonotic opportunistic bacterial pathogen that is ubiquitous in diverse environments, including water, animal-related food, the surface of medical instruments, and sewage systems in hospitals [1,2]. It carries a large variety of virulence factors and can cause bacteremia, ventilator-associated pneumonia, cystic fibrosis, and chronic obstructive pulmonary disease. It has the ability to form biofilms and attach to the surface of medical instruments and food [1]. It can spread in healthcare settings from one person to another

GCA_024714245.1. The complete sequence of 18083286 has been submitted to GenBank under accession number CP110368.

**Funding:** Funding for study design, data collection, data generation, and publication costs was provided by the National Science and Natural Science Foundation of China (Grant agreement 31872486) awarded to Zhu LW. The authors declare that the research was conducted in the absence of any commercial or financial relationships that could be construed as a potential conflict of interest.

**Competing interests:** The authors have declared that no competing interests exist.

through contaminated hands or surfaces. It is easily disseminated within hospitals; it caused an estimated 32,600 infections among hospitalized patients and resulted in approximately 2,700 deaths in the United States according to the Threat Estimate 2019 report [3]. Clinically, *P. aeruginosa* infection is usually treated by antimicrobial therapy, and prolonged use of antibiotics to achieve bacterial cure is commonly practiced [4]. Resistance genes can be acquired through the transfer of mobile genetic elements (such as plasmids, transposons, and integrative and conjugative elements) among bacterial strains, resulting in the development of multidrug-resistant *P. aeruginosa* in chronically infected patients [5,6]. Carbapenems are the most important antibiotics for treating multidrug-resistant bacterial infections, but *P. aeruginosa* is currently also resistant to carbapenems due to the acquisition of carbapenemases, among other reasons, impeding treatment.

In the present study, two *P. aeruginosa* isolates (18081308 and 18083286) were obtained from two patients (urine and sputum samples, respectively) admitted to a public hospital (Changchun, China) in August 2018. Their whole genome sequences and molecular characteristics were determined. Both isolates belonged to the same multilocus sequence type (ST865) and serotype (O6). They both carried $bla_{IMP-1}$. Detailed genetic dissection was applied to a Tn*7*-like transposon carrying $bla_{IMP-1}$ to display its genetic environment. The data presented here provide a deeper understanding of drug resistance gene acquisition in *P. aeruginosa* from a genomic and bioinformatic point of view.

## Materials and methods

### Bacterial isolation and identification

In the present case, a 72-year-old man (patient A) was admitted with cardiovascular disease (CVD) in August 13, 2018. Fourteen days later, another patient (patient B), a 56-year-old man was admitted with respiratory disease. Strains 18083286 and 18081308 were isolated from the sputum (patient A) and urine specimens (patient B) of the patients. The species was determined based on the partial sequence of the 16S rRNA gene [7].

Minimum inhibitory concentrations (MICs) of amikacin, gentamicin, meropenem, imipenem, cefazolin, ceftazidime, cefotaxime, cefepime, aztreonam, ampicillin, piperacillin, amoxicillin-clavulanate, ampicillin-sulbactam, piperacillin-tazobactam, trimethoprim-sulfamethoxazole, chloramphenicol, ciprofloxacin, levofloxacin, moxifloxacin, and tetracycline against strains 18083286 and 18081308 were tested by BD Phoenix-100, using *Escherichia coli* ATCC25922 as a control. Drug resistance and sensitivity were judged based on the Clinical and Laboratory Standards Institute guidelines (2019).

### Next-generation sequencing, sequence assembly and annotation

Bacterial genomic DNA was extracted from strains 18081308 and 18083286 using the Ultra-Clean Microbial Kit and sequenced using an Illumina NovaSeq PE150 platform. Trimmomatic V10 was used to remove the PCR adapters and low-quality reads, and SPAdes (http://cab.spbu.ru/software/spades/) was used for sequence assembly [8]. Precise species identification was performed by pairwise average nucleotide identity (ANI) (http://www.ezbiocloud.net/tools/ani) analysis between genome sequences and the *P. aeruginosa* reference genome PAO1 (GenBank ID: NC_002516.2). An ≥95% ANI cut-off was used to define bacterial species [9]. PAst (https://cge.food.dtu.dk/services/PAst/) was used to perform serotyping. Multilocus sequence types (STs) were obtained by uploading their genomes, including the seven conserved housekeeping genes *acsA*, *aroE*, *gtaA*, *mutL*, *nuoD*, *ppsA*, and *trpE*, to pubMLST (https://pubmlst.org/). Online databases, including CARD [10] (https://card.mcmaster.ca/)

and ResFinder 4.0 [11] (https://cge.cbs.dtu.dk/services/ResFinder/), were used to identify resistance genes.

### Single molecule real-time sequencing, annotation and comparison

The nucleotide identity between strains 18083286 and 18081308 was evaluated using ANI. Due to the discontinuity and incompleteness of the next-generation sequencing (NGS) results, there was an interference effect on the analysis of nucleobase absences. In this study, strain 18083286 was randomly selected to undergo another DNA extraction step, and the newly obtained DNA was subsequently single molecule real-time (SMRT) sequenced using a PacBio RSII sequencer. Based on the aforementioned sequencing data, Canu software (version 2.0) was utilized for genome assembly from reads, yielding initial assembly results that reflect the genomic status of the sample. Subsequently, Rcon software (version 1.4.13) was employed for three rounds of error correction based on third-generation sequencing data, followed by three rounds of Pilon software (version 1.22) error correction using second-generation reads, resulting in the final assembly outcome. Then, the sequenced DNA was annotated to identified mobile genetic elements (MGEs), and the MGE sequences were used to generate linear alignment maps with other sequences from the same family in GenBank. RAST *2.0* [12] and BLASTP/BLASTN [13] searches were conducted to predict open reading frames (ORFs). The CRAD [10] and ResFinder 4.0 [11] databases were used to identify drug resistance genes, again. ISfinder [14] (https://www-is.biotoul.fr/; last database update 2021-9-21), TnCentral (https://tncentral.ncc.unesp.br), INTEGRALL (http://integrall.bio.ua.pt/) [15], and ICEberg 2.0 (http://db-mml.sjtu.edu.cn/ICEberg/) [16] were used to identify mobile elements. Pairwise sequence comparisons were carried out by BLASTN. Gene organization diagrams were drawn by Inkscape 1.0 (http://inkscape.org/en/).

### Nucleotide sequence accession numbers

The contig sequences of strains 18081308 and 18083286 have been submitted to GenBank under accession numbers GCA_024718375.1 and GCA_024714245.1. The complete sequence of 18083286 has been submitted to GenBank under accession number CP110368.

### Results and discussion

Strains 18083286 and 18081308 were identified as *P. aeruginosa* by the BD Phoenix-100 identification system and based on the 16S rRNA gene. Table 1 shows the drug resistance spectrum of strain 18083286, which was consistent with that of strain 18081308.After Illumina NovaSeq PE150 sequencing (basic information about the Illumina sequencing results is provided in Table 2), it was found that their ANI values were more than 95% with the reference strain *P. aeruginosa* PAO1 (GenBank ID: NC_002516.2), and they were confirmed to be *P. aeruginosa* (ANI values of *P. aeruginosa* 18083286 and 18081308 are provided in S1 Table).

Both isolates belonged to the same multilocus sequence type (ST865) and serotype (O6) based on MLST and PAst screening. There are 20 serotypes of *P. aeruginosa*, of which serotype O6 is one of the most common [17]. ST865 is not a pandemic clonal group. Until 2022, the MLST database contained a total of four strains of *P. aeruginosa* ST865: strains 18081308 and 18083286 isolated in this study, strain AZPAE14882 of unknown origin, and strain AUS151 isolated from soft tissue in Australia in 2008.

They were resistant to many antibiotics in addition to imipenem, including: gentamicin, meropenem, cefazolin, ceftazidime, cefotaxime, cefepime, ampicillin, amoxicillin-clavulanate, ampicillin-sulbactam, trimethoprim-sulfamethoxazole, chloramphenicol, and tetracycline. The results are summarized in Table 1. Aminoglycoside resistance genes (*aac(6")-II*, *aac(3)-*

**Table 1. Antimicrobial susceptibility of strain 18083286.**

| Antimicrobial type | Antimicrobial | MIC (µg/mL)[a] | SIR[b] |
|---|---|---|---|
| Aminoglycoside | Amikacin | 16 | S |
| | Gentamicin | >8 | R |
| β-lactam | Imipenem | 8 | R |
| | Meropenem | 8 | R |
| | Cefazolin | >16 | R |
| | Ceftazidime | >16 | R |
| | Cefotaxime | >32 | R |
| | Cefepime | >16 | R |
| | Aztreonam | 8 | S |
| | Ampicillin | >16 | R |
| | Piperacillin | 16 | S |
| | Amoxicillin-clavulanate | >16/8 | R |
| | Ampicillin-sulbactam | >16/8 | R |
| | Piperacillin-tazobactam | 32/4 | I |
| Colistin | Colistin | 1 | NA |
| Sulfonamide | Trimethoprim-sulfamethoxazole | 2/38 | R |
| Chloramphenicol | Chloramphenicol | >16 | R |
| Quinolones | Ciprofloxacin | ≤0.5 | S |
| | Levofloxacin | ≤1 | S |
| | Moxifloxacin | 4 | NA |
| Tetracycline | Tetracycline | 8 | R |

[a]MIC, minimum inhibitory concentration.

[b]SIR, Susceptible (S), intermediate (I), resistant (R).

NA, not applicable.

Note: Strain 18081308 has the same resistance profile as 18083286.

*IId*, and *aph(3")-IIb*), an amphenicol resistance gene (*catB7*), β-lactam resistance genes ($bla_{OXA-486}$, $bla_{IMP-1}$, and $bla_{PAO}$), and a fosfomycin resistance gene (*fosA*) were identified by ResFinder in these strains.

Since the two strains had the same resistance profile, ST type, serotype, acquired resistance genes, and an ANI value of 99.99% (ANI values are provided in S1 Table) [18], one of the two strains was randomly selected for genetic environment analysis of the carbapenem resistance gene $bla_{IMP-1}$. SMRT sequencing (basic information about SMRT sequencing results is provided in Table 2) showed that the chromosome of strain 18083286 was 6.4 Mb and its GC

**Table 2. Basic information about bacterial sequencing results.**

| Strain name | Sequence size | Number of contigs | GC content (%) | Shortest contig size | Median sequence size | Mean sequence size | Longest contig size | N50 value | L50 value | Sequencing method |
|---|---|---|---|---|---|---|---|---|---|---|
| 18081308 | 6,398,311 | 29 | 66.4 | 1,174 | 109,405 | 220,631.4 | 904,406 | 424,532 | 6 | Illumina NovaSeq PE150 |
| 18083286 | 6,400,530 | 30 | 66.4 | 1,051 | 94,875 | 213,351.0 | 904,406 | 444,607 | 5 | Illumina NovaSeq PE150 |
| 18083286 | 6,433,752 | 1 | 66.4 | 6,433,752 | 6,433,752 | 6,433,752.0 | 6,433,752 | NA | 1 | Single-molecule real-time |

NA, not applicable.

content was 66.3%; no plasmid was detected. $bla_{IMP-1}$ was located in a Tn7-like transposon in the bacterial chromosome.

A 37.53-kb transposon was inserted at a *glmS* (glucosamine-fructose-6-phosphate amino-transferase) site in the chromosome of *P. aeruginosa* 18083286. It had a complete set of Tn7-family core transposon-encoded proteins (TnsABCDE), but with very low levels of nucleotide identity with Tn7 counterparts. This structure had the closest phylogenetic relationship with Tn*6411* (a Tn7-like family transposon) in *P. aeruginosa* 12939 (GenBank ID: CP024477.1; coverage: 100%, identity: 100%); thus, this Tn7-like transposon was identified as Tn*6411*. It was first discovered in a *P. aeruginosa* strain from China in 2018 [19].

Until October 2022, only 10 transposons with the same TnsA as Tn*6411* were indexed in GenBank (Table 3 shows strain information). Among them, strains 12939, 18083286, and HB2011305RE, and plasmid p201330 carried Tn*6411*, and others carried its derived structures, named Tn6*411*-like. They mainly included *P. aeruginosa* from China, but *Acinetobacter spp.* from Sydney, Australia, and *P. aeruginosa* from India were also found. Except for Tn*6411*$_{18083286}$, Tn*6411*-like$_{SE5430}$, and Tn*6411*-like$_{PA34}$ transposons, the others contained 20-bp TnsB-binding sites plus 26 bp inverted repeats (IRs), which were terminal flanking regions (S2 Table). They contained complete TnsABC+TnsD/E proteins, which were encoded by genes inserted into attTn7 or plasmids capable of transfer between bacteria. Except for Tn*6411*-like$_{PA34}$ (Indian), all others carried a truncated *aacC2-tmrB* region. The intact structure (IS*26-aacC2-tmrB* region remnant-$bla_{TEM-1}$) was found in pEl15573 [20,21]. It was derived from transposon Tn2. The structure (IS*26-aacC2-tmrB* region remnant) was found in the IncR/IncP6 fusion plasmid pCRE3-KPC carried by *Citrobacter braakii* [22]. The truncated *aacC2-tmrB* region identified in this study was likely to be an intact structure from which first $bla_{TEM-1}$ and then IS*26* was deleted.

All ten Tn*6411* transposons (until February 2022) listed in GenBank are shown in Fig 1. An integron carrying $bla_{IMP-1}$ (a carbapenem resistance gene) and *aac(6")-II* (an aminoglycoside resistance gene), named In992, was inserted between *pinR* (encoding a DNA site-specific recombinase) and a gene (encoding a methyltransferase domain protein) that serves the backbone of Tn*6411*$_{12939}$ (Beijing, China), Tn*6411*$_{18083286}$ (Changchun, China), Tn*6411*$_{HB2011305RE}$ (Changchun, China), and Tn*6411*$_{p201330}$ (Changchun, China). One DR copy (ATGCCCGC) of In992 was found upstream of *pinR* (Tn*6411*-like$_{P8W}$, Tn*6411*-like$_{P9W}$, and Tn*6411*-like$_{SE5430}$), which enabled the insertion of In992, resulting in a bilateral DR sequence (ATGCCCGC). Tn*6411* was carried by other strains that do not carry In992 or other integrons. The truncated

**Table 3. The information of strains carrying Tn*6411* and its derived structures.**

| Strain | Source | Species | Location[a] | Year | Country | Size (Mb) | GC content (%) | Assembly ID |
|---|---|---|---|---|---|---|---|---|
| p201330-IMP | Homo sapiens | *P. eruginosa* | P | 2013 | Dalian, China | 0.17 | 58.5 | MN961671.1 |
| 12939 | Homo sapiens, sputum | *P. eruginosa* | C | 2013 | Beijing, China | 6.62 | 66.2 | CP024477.1 |
| P8W | Homo sapiens, burn wound | *P. eruginosa* | C | 2018 | Tianjin, China | 7.19 | 65.9 | CP081477.1 |
| P9W | Homo sapiens, burn wound | *P. eruginosa* | C | 2018 | Tianjin, China | 7.18 | 65.9 | CP081202.1 |
| pE47 | Homo sapiens | *A. baumannii* | P | 2013 | Sydney, Australia | 0.32 | 40.8 | CP042557.1 |
| pWM98B | Homo sapiens, sputum | *A. nosocomialis* | P | 1998 | Sydney, Australia | 0.26 | 41.4 | MT742183.1 |
| pMKPA34-2 | Homo sapiens, sputum | *P. eruginosa* | P | 1997 | India | 0.03 | 61 | MH547561.1 |
| 18083286 | Homo sapiens, sputum | *P. eruginosa* | C | 2018 | Changchun, China | 6.43 | 66.3 | CP110368 |
| HB2011305RE | Homo sapiens | *P. eruginosa* | C | 2011 | Hebei, China | 6.84 | 66 | CP054787.1 |
| SE5430 | Homo sapiens | *P. eruginosa* | C | 2012 | Suzhou, China | 6.83 | 65.9 | CP054791.1 |

a. P, plasmid; C, chromosome.

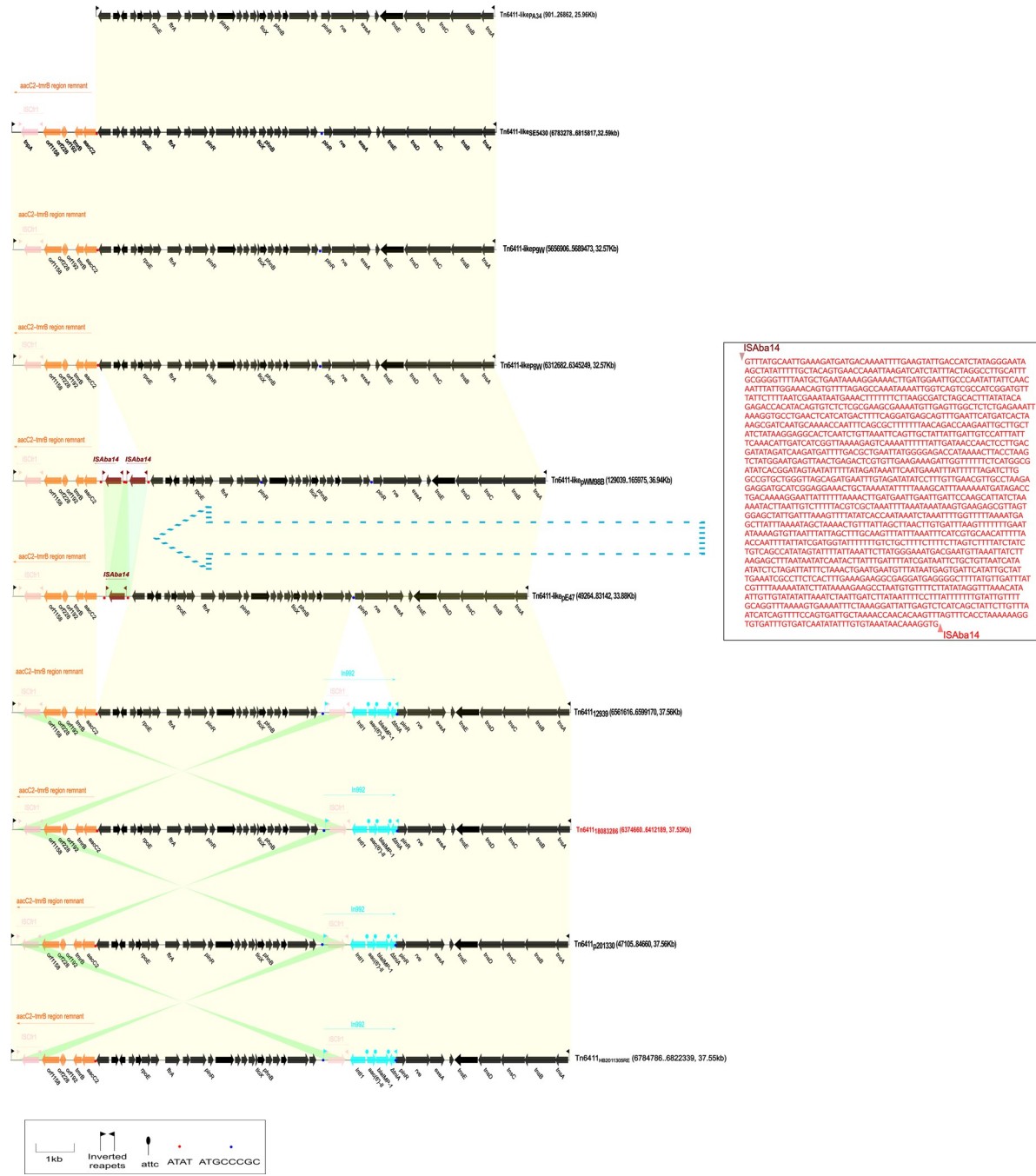

**Fig 1. Linear alignment map of Tn*6411* and its derived structures.** The backbone region is shown in black, In992 is shown in light blue, the *aacC2-tmrB* region remnant is shown in orange, IS*Aba14* is shown in brownish red, and IS*Cfr1* is shown in pink. The shaded region represents a region with >90% nucleotide identity. All transposons encoded complete TnsABC+TnsD/E proteins. Except for Tn*6411*-like_PA34, all others carried a truncated *aacC2-tmrB* region remnant. An integron named In992 carrying *bla*_IMP-1 and *aac(6")-II* was inserted between *pinR* and a gene encoding a methyltransferase domain protein from the backbone of Tn*6411*_12939, Tn*6411*_18083286, Tn*6411*_HB2011305RE, Tn*6411*_p201330, and Tn*6411* carried by other strains that do not carry In992 or other integrons. The truncated Tn*402* transposition module in In992 had undergone the deletion of the partial TniA sequence. Although Tn*6411*-like_pE47 and Tn*6411*-like_pWM98B from *A. baumannii* and *A. nosocomialis* did not carry In992, one or two copies of IS*Aba14* were inserted downstream of the *aacC2-tmrB* region remnant. Tn*6411*-like_pE47 had a one-copy IS*Aba14* difference from Tn*6411*-like_pWM98B in addition to a 1777-bp sequence difference.

Tn*402* transposition module in In992 had undergone the deletion of the partial TniA sequence and the complete TniBQR sequence, resulting in the loss of its self-transfer capability.

One or two copies of IS*Aba14* were inserted upstream of the *aacC2-tmrB* region remnant, although Tn*6411*-like<sub>pE47</sub> and Tn*6411*-like<sub>pWM98B</sub> from *Acinetobacter baumannii* (Sydney, Australia) and *Acinetobacter nosocomialis* (Sydney, Australia) did not carry In992. When one copy of IS*Aba14* and the 1777-bp neighbor base sequence were lost and the other copy of IS*Aba14* was retained, a Tn*6411*-like<sub>pWM98B</sub> (Tn*6411*-*aacC2-tmrB* region remnant-IS*Aba14*-IS*Aba14*) changed into Tn*6411*-like<sub>pE47</sub> (Tn*6411*-*aacC2-tmrB* region remnant-IS*Aba14*) (Fig 1).

Tn*6411*-like<sub>P8W</sub>, Tn*6411*-like<sub>P9W</sub>, Tn*6411*-like<sub>SE5430</sub>, and Tn*6411*-like<sub>PA34</sub> did not have an accessory module (the sequence of Tn*6411*-like<sub>PA34</sub> was discontinuous and incomplete, and no analysis was conducted). One copy of the DR sequence (ATAT) of IS*Aba14* was found upstream of the *aacC2-tmrB* region remnant of Tn*6411*-like<sub>P8W</sub>, Tn*6411*-like<sub>P9W</sub>, and Tn*6411*-like<sub>SE5430</sub>, which enabled double copy of the IS*Aba14* sequence in the same direction to insert its Tn*6411*-like structure (Fig 1).

Compared with Tn*6411*-like<sub>SE5430</sub>, Tn*6411*-like<sub>P8W</sub> and Tn*6411*-like<sub>P9W</sub> lacked a 26-bp sequence downstream of the *aacC2-tmrB* region remnant, but the IRL of Tn*6411*-like<sub>SE5430</sub> lacked a 14-bp sequence. TnsB binding site 1 had an 8-bp deletion as Tn*6411*<sub>18083286</sub>, but the difference was that TnsB binding sites 2 and 3 of Tn*6411*-like<sub>SE5430</sub> were also missing.

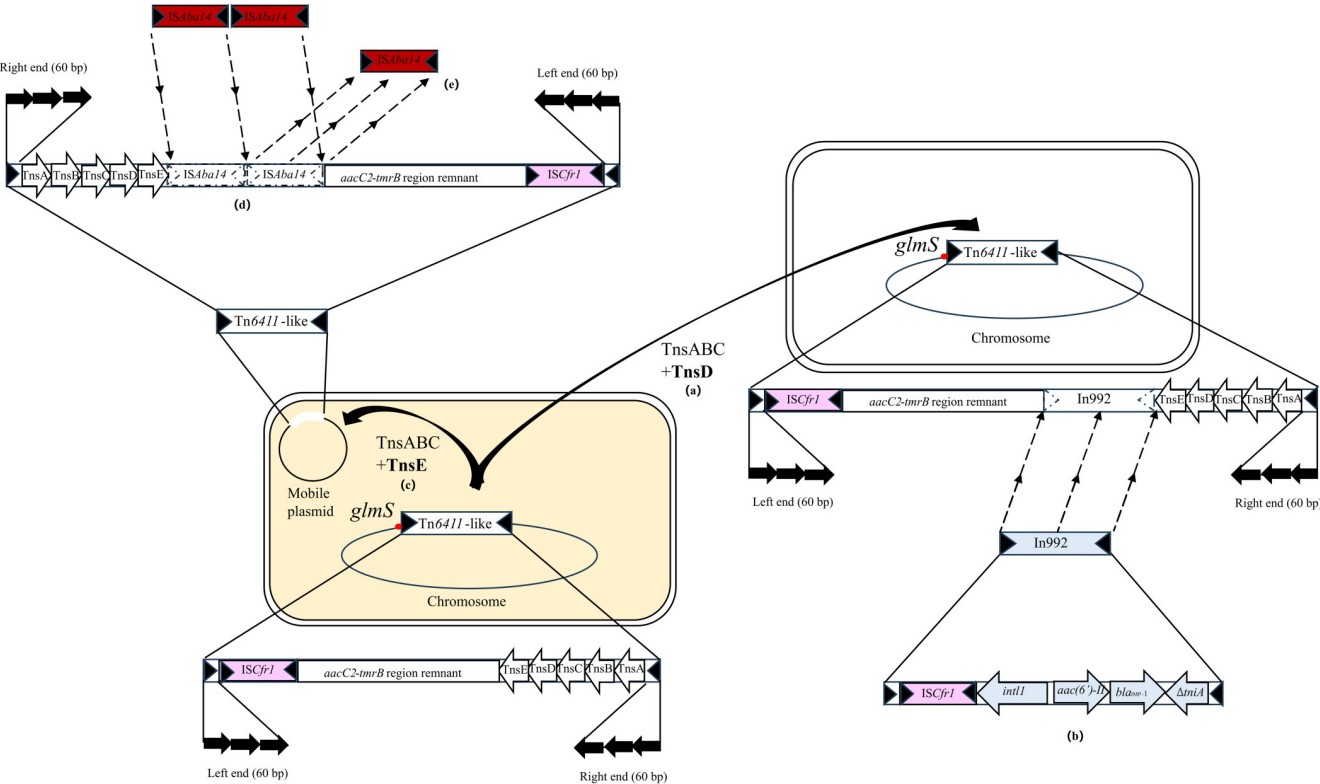

**Fig 2. The potential formation process of Tn*6411* and its derived structures.** The figure shows the *aacC2-tmrB* region remnant-Tn*6411* backbone as a local structure. IS*Cfr1* is shown in pink, attTn*7* (*glmS*) is shown as a red dot, In992 is shown in light blue, and IS*Aba14* is shown in brownish red. **(a)** A Tn*6411*-like transposon (*aacC2-tmrB* region remnant-Tn*6411*backbone) was localized in the chromosome by TnsD; **(b)** In992 carrying *bla*<sub>IMP-1</sub> and *aac(6")-II* was inserted into the Tn*6411* backbone. The truncated Tn*402* transposition module in In992 had undergone the deletion of the partial TniA sequence and complete TniBQR sequence and lost its self-transfer capability, making it stable, forming Tn*6411*. **(c)** Tn*6411*-like (*aacC2-tmrB* region remnant-Tn*6411*) was localized in a plasmid by TnsE for horizontal transmission. **(d)** Two copies of IS*Aba14* were inserted into the Tn*6411* backbone. **(e)** One copy of IS*Aba14* and a 1777-bp neighbor base sequence were lost, and the other copy of IS*Aba14* was retained.

Therefore, the formation of the structure of Tn*6411*-like$_{P8W}$ and Tn*6411*-like$_{P9W}$ might have occurred before the formation of Tn*6411*-like$_{SE5430}$ (S2 Table).

Genetic sequence analysis revealed the potential structure change and transfer of the Tn*6411* transposons. The *aacC2-tmrB* region remnant-Tn*6411* backbone was the earliest structure, and then in the evolutionary process, the element might form two evolutionary modes; one was localized in chromosome by TnsD for vertical transmission within species, and the other was localized in plasmid by TnsE for horizontal transmission between different species (Fig 2).

In conclusion, genetic sequence analysis suggested that Tn*6411* transposons could act as vectors and capture type 1 integrons containing *bla*$_{IMP-1}$. Although the Tn*6411* transposons sequences were commonly situated within the chromosome of *P. aeruginosa*, they had also been identified in the plasmids carried by *Acinetobacter spp.*. The absence of experimental verification for horizontal gene transfer also presented constraints on this research, rendering its transferability uncertain. The presence of Tn*6411* sequences in various species indicated that there should be a closer monitoring of and investigation into the conditions under which this transfer occurs.

## Supporting information

**S1 Table. The ANI value of *P. aeruginosa* in this study.**
(XLSX)

**S2 Table. The sequence of TnsB-binding sites and IRs.**
(XLSX)

## Acknowledgments

We are grateful to the members of the China-Japan Union Hospital, Jilin University.

## Author Contributions

**Investigation:** Gejin Lu, Jie Jing, Shiwen Sun, Yang Sun, Xue Ji, Bowen Jiang, Yongjie Wang.

**Methodology:** Jiayao Guan, Chuanfang Zhao.

**Visualization:** Jingyi Guo.

**Writing – original draft:** Lin Zheng, Zixian Wang.

**Writing – review & editing:** Lingwei Zhu, Xuejun Guo.

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
