## [Decision Letter · Decision Letter 0]

12 Apr 2024

PONE-D-24-01335Potential transferability and evolution of Tn6411 transposons carrying the blaIMP-1 gene in Pseudomonas aeruginosaPLOS ONE

Dear Dr. Guo,

Thank you for submitting your manuscript to PLOS ONE. After careful consideration, we feel that it has merit but does not fully meet PLOS ONE’s publication criteria as it currently stands. Therefore, we invite you to submit a revised version of the manuscript that addresses the points raised during the review process.

We look forward to receiving your revised manuscript.

Kind regards,

Mohamed O Ahmed, Ph.D

Academic Editor

PLOS ONE

Journal Requirements:

"National Science and Natural Science Foundation of China (Grant agreement 31872486)"

4. Please note that your Data Availability Statement is currently missing a direct link to access each database. If your manuscript is accepted for publication, you will be asked to provide these details on a very short timeline. We therefore suggest that you provide this information now, though we will not hold up the peer review process if you are unable.

Reviewers' comments:

Reviewer's Responses to Questions

**Comments to the Author**

1. Is the manuscript technically sound, and do the data support the conclusions?

Reviewer #1: Yes

Reviewer #2: Partly

2. Has the statistical analysis been performed appropriately and rigorously? 

Reviewer #1: N/A

Reviewer #2: N/A

3. Have the authors made all data underlying the findings in their manuscript fully available?

Reviewer #1: Yes

Reviewer #2: Yes

4. Is the manuscript presented in an intelligible fashion and written in standard English?

Reviewer #1: Yes

Reviewer #2: Yes

5. Review Comments to the Author

Reviewer #1: The manuscript by Wang and cols. analyzes the genetic detection of transposon Tn6411 flanking the blaIMP-1 carbapenem-resistant gene in two Pseudomonas aeruginosa of clinical oring. This manuscript is relevant because it reports the evolution and potential dissemination of Tn6411 in clones of P. aeruginosa not typically reported, considered not pandemic, which highlights its role in spreading last-resort antibiotics.

However, I have some recommendations that could improve the manuscript:

1. The quality of the two figures must be improved.

2. Line 81. Please include the version (year) of the LCSI guidelines.

3. Line 109-111. Why did the authors not include the OD600 as a normalization tool for the number of cells in the conjugation experiments? Why was a P. aeruginosa strain not used as a recipient instead of an E. coli? Please add more information about it in the discussion section.

4. Line 150-151. Why was strain 18083286 included in the conjugation experiments even though "no plasmid was detected"? Maybe the order of the experiments should be revised.

Reviewer #2: The paper “Potential transferability and evolution of Tn6411 transposons carrying the blaIMP-1 gene in Pseudomonas aeruginosa” is interesting. Describes a transposon that allows the mobility of resistance genes such as blaIMP-1. However, major corrections must be made to make it clear.

You must change Figure 1 because it cannot be seen.

Table 1 only describes the resistance profile of strain 18083286. If strain 18081308 has the same resistance profile, it should be said in the footer of the table or the title, and if not, the resistance profile of this strain should also be included.

The authors describe that this transposon carrying blaIMP-1 is found in one of the strains on the chromosome and in the other on a plasmid. However, they do not describe the plasmid carrying the transposon by sequencing, which would be important to see if it coincides with the results obteined in the assay of the conjugation.

In methods and results, the authors do not specify the strain used. The article, in general, describes one of the strains, and the question remains whether the other strain had the same thing or if there are changes, so it should include both strains.

The authors show that two methodologies were used for sequencing strain 18083286; however, they do not say if it made a hybrid assembly of the sequences.

6. PLOS authors have the option to publish the peer review history of their article (what does this mean?). If published, this will include your full peer review and any attached files.

Reviewer #1: **Yes: **Gerardo Cortés-Cortés

Reviewer #2: No

---

## [Author Response · Author response to Decision Letter 0]

2 May 2024

We are greatly appreciated to editor's contribution to this article. We have completed the modification of the paper format as required, and at the same time we have sought native English speakers to help us polish the language in this article. We have also released the relevant sequences in GenBank for readers' reference.

Reviewer #1: The manuscript by Wang and cols. analyzes the genetic detection of transposon Tn6411 flanking the blaIMP-1 carbapenem-resistant gene in two Pseudomonas aeruginosa of clinical origin. This manuscript is relevant because it reports the evolution and potential dissemination of Tn6411 in clones of P. aeruginosa not typically reported, considered not pandemic, which highlights its role in spreading last-resort antibiotics. However, I have some recommendations that could improve the manuscript: 

Response: Thank Reviewer 1;

we learned a lot from the valuable advice and suggestions. We have thoroughly revised our manuscript. The issues have been addressed point-by-point. Please see our responses below.

1. The quality of the two figures must be improved.

Response: Revised (Fig 1 and Fig 2);

The clarity of the image is directly correlated with the reader's perception. Currently, we have enhanced the picture's clarity using photoshop software by increasing the original 300 dpi to 600 dpi and converting the output from “png” to “tif” format. As a result, there has been a significant improvement in clarity.

2. Line 81. Please include the version (year) of the CLSI guidelines.

Response: Added (Line 84-86): “Drug resistance and sensitivity were judged based on the Clinical and Laboratory Standards Institute guidelines (2019).”

Thank you for the expert's reminder. We overlooked this aspect, as CLSI undergoes minor changes with each update, so it is important to specify the version we are referring to. Therefore, we have added the CLSI version (2019) for reader reference.

3. Line 109-111. Why did the authors not include the OD600 as a normalization tool for the number of cells in the conjugation experiments? Why was a P. aeruginosa strain not used as a recipient instead of an E. coli? Please add more information about it in the discussion section.

Response: Added (Line 138-139): “The flat colony counting method was used to ensure an equal 1:1 mixture of each strain with E. coli J53.”

The description of "equal volume" in this article is overly general, and a normalization method should be employed to describe. The OD600 method is used to estimate the number of cells by measuring the optical density of the cell suspension at a wavelength of 600 nanometers. However, it lacks the ability to differentiate between living and dead cells, and variations in light absorption properties among different cell types or within the same cell at different growth stages, which can lead to inaccuracies in OD600 measurements. This limitation is particularly significant in experiments requiring precise cell counting. Therefore, this study employed the technique of flat colony counting method to ascertain if the quantities were in an equal ratio. Currently, this aspect of the experimental method has been refined.

There were several uncertainties regarding the experimental conditions when using non-E. coli J53 as recipient cells for conjugation. Therefore, the E. coli J53 strain, which is commonly used for conjugation experiments, was chosen as the recipient strain in this study.

4. Line 150-151. Why was strain 18083286 included in the conjugation experiments even though "no plasmid was detected"? Maybe the order of the experiments should be revised.

Response:

In this study, we initially identified high similarity between the nucleic acid sequences of strain 18083286 and 18081308 using next-generation sequencing. Consequently, only strain 18083286 was randomly sequenced for the single molecule real-time sequencing. Despite not detecting plasmids in the bacterium through sequencing, it was important to note that current sequencing technology may easily overlook low-copy number plasmids. Therefore, the presence of bacteria carrying low-copy plasmids cannot be ruled out by sequencing alone.

Comparison with the GenBank database revealed that Tn6411 sequences existed not only in Pseudomonas aeruginosa but also in Acinetobacter spp.. Subsequently, conjugation experiments were conducted to ascertain whether the blaIMP-1 gene in Tn6411 is transferable. Hence, we performed conjugation experiments after sequencing to validate the transferability of Tn6411-blaIMP-1.

Reviewer #2: The paper “Potential transferability and evolution of Tn6411 transposons carrying the blaIMP-1 gene in Pseudomonas aeruginosa” is interesting. Describes a transposon that allows the mobility of resistance genes such as blaIMP. However, major corrections must be made to make it clear. 

Response: Thank Reviewer 2’s suggestions. 

We have revised our manuscript point-by-point. Please see the following response for details:

1. You must change Figure 1 because it cannot be seen. 

Response: Revised (Fig 1 and Fig 2);

The clarity of the image was directly related to how the reader seen it. We recently improved the picture's clarity using photoshop software, increasing the original 300 dpi to 600 dpi and changing the output from “png” to “tif” format. As a result, there had been a significant improvement in clarity.

2. Table 1 only describes the resistance profile of strain 18083286. If strain 18081308 has the same resistance profile, it should be said in the footer of the table or the title, and if not, the resistance profile of this strain should also be included. 

Response: Revised (Line 153-155): “Table 1 shows the drug resistance spectrum of strain 18083286, which was consistent with that of strain 18081308.”

Thank you once again for the expert's reminder. The drug resistance information of strains 18083286 and 18081308 is indicated consistent in the article (Line 174), but only the drug resistance information of strain 18083286 is included in the Table 1. This omission may lead to confusion among some readers. As suggested by experts, when referring to Table 1, it is explained that the resistance of the two is consistent. 

3. The authors describe that this transposon carrying blaIMP-1 is found in one of the strains on the chromosome and in the other on a plasmid. However, they do not describe the plasmid carrying the transposon by sequencing, which would be important to see if it coincides with the results obtained in the assay of the conjugation.

Response: 

The strain Tn641118083286 identified in this study is chromosomally located, while the plasmid-mediated Tn6411-likep201330-IMP, Tn6411-likepE47, Tn6411pWM98B, and Tn6411pMKPA34-2 are carried by the selected strains in GenBank (Line 190-191). Whether conjugal transfer events can occur has not been reported.

4. In methods and results, the authors do not specify the strain used. The article, in general, describes one of the strains, and the question remains whether the other strain had the same thing or if there are changes, so it should include both strains. 

Response: Revised (Line 79-139; Line 153-155);

The absence of strain labeling poses a challenge for readers in comprehending this article. Currently, we have provided specific details about the strains used to enhance readers' understanding of this study.

The nucleotide identity between strains 18083286 and 18081308 was evaluated using ANI. Due to the discontinuity and incompleteness of the next-generation sequencing (NGS) results, there was an interference effect on the analysis of nucleobase absences. In this study, strain 18083286 was randomly selected to undergo another DNA extraction step, and the newly obtained DNA was subsequently single molecule real-time (SMRT) sequenced using a PacBio RSII sequencer. Therefore, the phenotype experiment in this paper was analyzed between strains 18083286 and 18083286. Subsequent sequence analysis was carried out between strain 18083286 and the GenBank database.

5. The authors show that two methodologies were used for sequencing strain 18083286; however, they do not say if it made a hybrid assembly of the sequences. 

Response: Revised (Line 94-128)

We initially performed next-generation sequencing on 18083286 to preliminarily assess its nucleic acid sequence identity with strain 18081308. Due to the high-level similarity of nucleic acid sequence, presence of drug-resistance genes and phenotypes between 18083286 and 18081308, we proceeded single molecule real-time sequencing on a randomly selected strain. As a result, both next-generation and real-time sequencing of this bacterium were conducted as non-mixed sequencing. The experimental methods of next-generation sequencing and single molecule real-time sequencing analysis have been described separately to ensure clear expression.

---

## [Decision Letter · Decision Letter 1]

27 May 2024

PONE-D-24-01335R1Potential transferability and evolution of Tn6411 transposons carrying the blaIMP-1 gene in Pseudomonas aeruginosaPLOS ONE

Dear Dr. Guo,

Thank you for submitting your manuscript to PLOS ONE. After careful consideration, we feel that it has merit but does not fully meet PLOS ONE’s publication criteria as it currently stands. Therefore, we invite you to submit a revised version of the manuscript that addresses the points raised during the review process.

We look forward to receiving your revised manuscript.

Kind regards,

Mohamed O Ahmed

Academic Editor

PLOS ONE

Reviewers' comments:

Reviewer's Responses to Questions

**Comments to the Author**

1. If the authors have adequately addressed your comments raised in a previous round of review and you feel that this manuscript is now acceptable for publication, you may indicate that here to bypass the “Comments to the Author” section, enter your conflict of interest statement in the “Confidential to Editor” section, and submit your "Accept" recommendation.

Reviewer #1: All comments have been addressed

Reviewer #2: (No Response)

2. Is the manuscript technically sound, and do the data support the conclusions?

Reviewer #1: Yes

Reviewer #2: Partly

3. Has the statistical analysis been performed appropriately and rigorously? 

Reviewer #1: N/A

Reviewer #2: N/A

4. Have the authors made all data underlying the findings in their manuscript fully available?

Reviewer #1: Yes

Reviewer #2: Yes

5. Is the manuscript presented in an intelligible fashion and written in standard English?

Reviewer #1: Yes

Reviewer #2: Yes

6. Review Comments to the Author

Reviewer #1: Dear authors,

Thanks for addressing my comments.

I have some minor suggestions that I consider important to clarify:

1. I partially agree that in this case, "the clarity of the image is directly correlated with the reader's perception". However, once the figures meet the standards indicated in the guide for authors, it should be fine, and the production team will say something in case not.

2. Regarding the conjugation experiments, it is not clear to me why you wanted to validate the transferability of Tn6411-blaIMP-1 by conjugation in a plasmid-free strain. Why don't you evaluate this transferability by transformation instead? which could provide more comprehensive results.

Reviewer #2: The article “Potential transferability and evolution of Tn6411 a transposons carrying the blaIMP-1 gene in Pseudomonas aeruginosa” describes a transposon that allows the mobility of resistance genes such as blaIMP-1. However, it remains unclear on the following points:

1. The authors do not answer whether they carried out a hybrid assembly for strain 18083286 because they carried out two sequencing by two different methods. In that case, they could carry out a hybrid assembly that would provide them with more information. They could be surer of your information. I suggest that it be carried out or clarify why it was carried out by two sequencing methodologies, both in methods and results.

2. Line 222. The authors describe the Tn6411 transposon through bioinformatic analysis and compare it with others reported in GenBank. However, they cannot say that they determined the evolution of this transposon since, with this analysis, they can only say that there are changes. Other studies must be done to know evolution, so the word evolution must be eliminated from the title.

3. Line 198. The authors say that the observed modifications make Tn6411 and Tn6411-like stable, which must be demonstrated experimentally. Like in line 229, they should remove “forming a stable structure when you capture the type1 integron with blaIMP-1 because bioinformatic analysis cannot determine."

4. Line 116. It is unclear why a conjugation test should be performed on strain 18083286, given that the bioinformatic analysis showed that Tn6411 with blaIMP-1 is located on the chromosome. We know that it will only be transferred vertically.

5. The authors say that the plasmid cannot be detected because they have a low copy number; however, it is not a factor for not observing it in the type of sequencing they carry out because, in these cases, libraries are created that in some way would help with this problem. On the other hand, they detect it in strain 18081308. However, they do not describe it, at least what refers to its conjugation machinery, to relate it to what was found in the conjugation assay. So, they must change their conclusion (line 233).

6. I continue to insist that the figures are of very poor quality and do not look properly. These must be clear to any reader.

7. The authors give the title “Potential transferability and evolution”; however, the only thing they do is a description of the transposon found in their strains and a comparison with other transposons reported in GenBank, so they cannot talk about evolution or Potential transferability. Only with a bioinformatic analysis, in addition to this not being adequately documented in the article

7. PLOS authors have the option to publish the peer review history of their article (what does this mean?). If published, this will include your full peer review and any attached files.

Reviewer #1: **Yes: **Gerardo Cortés-Cortés

Reviewer #2: No

---

## [Author Response · Author response to Decision Letter 1]

4 Jun 2024

Reviewer #1

1. I partially agree that in this case, "the clarity of the image is directly correlated with the reader's perception". However, once the figures meet the standards indicated in the guide for authors, it should be fine, and the production team will say something in case not.

Thank you for reviewer’s understanding. The figures have been adjusted based on “The guide for authors”, and it is possible that the journal may have compressed the figures when storing them. If the production team needs them in the future, we will be happy to provide assistance, including supplying the original figures.

2. Regarding the conjugation experiments, it is not clear to me why you wanted to validate the transferability of Tn6411-blaIMP-1 by conjugation in a plasmid-free strain. Why don't you evaluate this transferability by transformation instead? which could provide more comprehensive results.

Thank you for reviewer’s advice. Donor DNA and competent bacterial cells were needed for transformation instead. However, the strain 18083286 in this study did not contain plasmids, which made it challenging to extract DNA from the Tn6411 target fragment. The process of conducting adsorption experiments for free DNA without a plasmid-carrier has not been mature yet, adding to the uncertainty of the experiment. Therefore, this study didn’t conduct any transformation experiments.

We can also determine that it is only capable of vertical transmission on the chromosome by single molecule real-time (SMRT) sequencing and analyzing the structure of Tn6411. Therefore, the conjugation experiment in this study was not very meaningful and will be removed.

Reviewer #2

1. The authors do not answer whether they carried out a hybrid assembly for strain 18083286 because they carried out two sequencing by two different methods. In that case, they could carry out a hybrid assembly that would provide them with more information. They could be surer of your information. I suggest that it be carried out or clarify why it was carried out by two sequencing methodologies, both in methods and results.

Thank you for the reviewer’s advice. The reasons for conducting next generation sequencing (NGS) of the strains in this study are as follows:

(1) Due to the discontinuity and incompleteness of the NGS results, there was an interference effect on the analysis of nucleobase absences. To better understand the sequence structure, we conducted sequencing across single molecule real-time (SMRT) sequencing (Line: 106-110). Before proceeding with SMRT sequencing, in order to consider the cost, we initially used next generation sequencing to screen the strains;

(2) For the SMRT sequencing assembly, we utilized NGS calibration (Line: 110-116).

2. Line 222. The authors describe the Tn6411 transposon through bioinformatic analysis and compare it with others reported in GenBank. However, they cannot say that they determined the evolution of this transposon since, with this analysis, they can only say that there are changes. Other studies must be done to know evolution, so the word evolution must be eliminated from the title.

Thank you for the reviewer’s reminder. The title has been modified to “Comparative genomics of Tn6411 transposons carrying the blaIMP-1 gene in Pseudomonas aeruginosa”.

3. Line 198. The authors say that the observed modifications make Tn6411 and Tn6411-like stable, which must be demonstrated experimentally. Like in line 229, they should remove “forming a stable structure when you capture the type1 integron with blaIMP-1 because bioinformatic analysis cannot determine."

Thank you once again for the reviewer’s reminder. The bioinformatics focused on analyzing and predicting the function and structure of sequence, including mobile genetic elements (like integrons and transposons). According to the sequence perspective, the deletion of transposon unit sequences in this study could result in the mobility loss of Tn6411 and integrons (Line: 209-211). However, reviewer has emphasized that further experimental verification was needed to confirm whether they were stably present. Therefore, we removed speculations that have not been verified by experiments.

4. Line 116. It is unclear why a conjugation test should be performed on strain 18083286, given that the bioinformatic analysis showed that Tn6411 with blaIMP-1 is located on the chromosome. We know that it will only be transferred vertically.

We also thought that using single molecule real-time (SMRT) sequencing to analyze the structure of Tn6411 will show that it can only be passed down vertically on the chromosome. Therefore, the conjugation experiment in this study wasn't very useful and will be eliminated.

5. The authors say that the plasmid cannot be detected because they have a low copy number; however, it is not a factor for not observing it in the type of sequencing they carry out because, in these cases, libraries are created that in some way would help with this problem. On the other hand, they detect it in strain 18081308. However, they do not describe it, at least what refers to its conjugation machinery, to relate it to what was found in the conjugation assay. So, they must change their conclusion (line 233).

Thank you for reviewer’s clear explanation. 

Since strain 18081308 only underwent NGS, we were unable to determine whether it carried plasmids, so it was not mentioned in this study.

We have updated the conclusion by adjusting speculative statements about gene transfer that cannot be confirmed through bioinformatics analysis in this study (Line: 241-259).

6. I continue to insist that the figures are of very poor quality and do not look properly. These must be clear to any reader.

The figures have been adjusted according to "The guide for authors", and it's possible that the journal may have shrunk the figures when saving them. If the production team needs them in the future, we'd be happy to help, including providing the original figures.

7. The authors give the title “Potential transferability and evolution”; however, the only thing they do is a description of the transposon found in their strains and a comparison with other transposons reported in GenBank, so they cannot talk about evolution or Potential transferability. Only with a bioinformatic analysis, in addition to this not being adequately documented in the article

To better align with the research focus of this article, the title has been modified to “Comparative genomics of Tn6411 transposons carrying the blaIMP-1 gene in Pseudomonas aeruginosa”.

---

## [Decision Letter · Decision Letter 2]

18 Jun 2024

Comparative genomics of Tn6411 transposons carrying the blaIMP-1 gene in Pseudomonas aeruginosa

PONE-D-24-01335R2

Dear Guo,

We’re pleased to inform you that your manuscript has been judged scientifically suitable for publication and will be formally accepted for publication once it meets all outstanding technical requirements.

Kind regards,

Academic Editor

PLOS ONE

---

## [Editor Report · Acceptance letter]

28 Jun 2024

PONE-D-24-01335R2 

PLOS ONE

Dear Dr. Guo, 

I'm pleased to inform you that your manuscript has been deemed suitable for publication in PLOS ONE. Congratulations! Your manuscript is now being handed over to our production team.

Kind regards, 

on behalf of

Dr. Mohamed O Ahmed 

Academic Editor

PLOS ONE